# Efficient Exploration of Reward Functions in Inverse Reinforcement Learning via Bayesian Optimization

**Sreejith Balakrishnan, Quoc Phong Nguyen, Bryan Kian Hsiang Low, and Harold Soh**
Dept. of Computer Science, National University of Singapore, Republic of Singapore
{sreejith,qphong,lowkh,harold}@comp.nus.edu.sg

## Abstract

The problem of inverse reinforcement learning (IRL) is relevant to a variety of tasks including value alignment and robot learning from demonstration. Despite significant algorithmic contributions in recent years, IRL remains an ill-posed problem at its core; multiple reward functions coincide with the observed behavior and the actual reward function is not identifiable without prior knowledge or supplementary information. This paper presents an IRL framework called Bayesian optimization-IRL (BO-IRL) which identifies multiple solutions that are consistent with the expert demonstrations by efficiently exploring the reward function space. BO-IRL achieves this by utilizing Bayesian Optimization along with our newly proposed kernel that (a) projects the parameters of policy invariant reward functions to a single point in a latent space and (b) ensures nearby points in the latent space correspond to reward functions yielding similar likelihoods. This projection allows the use of standard stationary kernels in the latent space to capture the correlations present across the reward function space. Empirical results on synthetic and real-world environments (model-free and model-based) show that BO-IRL discovers multiple reward functions while minimizing the number of expensive exact policy optimizations.

## 1 Introduction

Inverse reinforcement learning (IRL) is the problem of inferring the reward function of a reinforcement learning (RL) agent from its observed behavior [1]. Despite wide-spread application (e.g., [1, 4, 5, 27]), IRL remains a challenging problem. A key difficulty is that IRL is ill-posed; typically, there exist many solutions (reward functions) for which a given behavior is optimal [2, 3, 29] and it is not possible to infer the true reward function from among these alternatives without additional information, such as prior knowledge or more informative demonstrations [9, 15].

Given the ill-posed nature of IRL, we adopt the perspective that an IRL algorithm should characterize the space of solutions rather than output a single answer. Indeed, there is often *no one* correct solution. Although this approach differs from traditional gradient-based IRL methods [38] and modern deep incarnations that converge to specific solutions in the reward function space (e.g., [12, 14]), it is not entirely unconventional. Previous approaches, notably Bayesian IRL (BIRL) [32], share this view and return a posterior distribution over possible reward functions. However, BIRL and other similar methods [25] are computationally expensive (often due to exact policy optimization steps) or suffer from issues such as overfitting [8].

In this paper, we pursue a novel approach to IRL by using Bayesian optimization (BO) [26] to minimize the negative log-likelihood (NLL) of the expert demonstrations with respect to reward functions. BO is specifically designed for optimizing expensive functions by strategically picking inputs to evaluate and appears to be a natural fit for this task. In addition to the samples procured, the Gaussian process (GP) regression used in BO returns additional information about the discovered

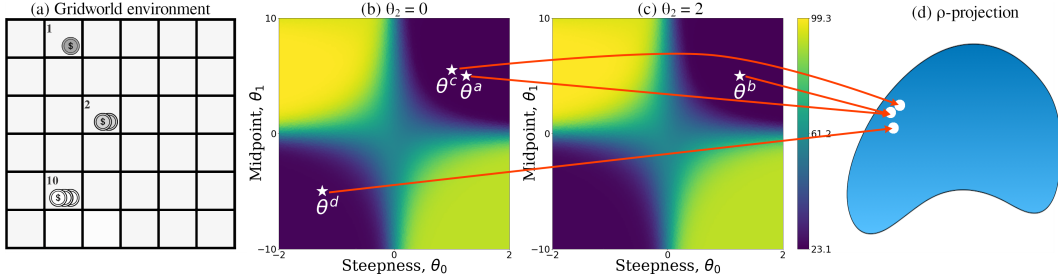

Figure 1: Our BO-IRL framework makes use of the $\rho$-projection that maps reward functions into a space where covariances can be ascertained using a standard stationary kernel. (a) Our running example of a $6 \times 6$ Gridworld example where the goal is to collect as many coins as possible. The reward function is modeled by a translated logistic function $R_{\boldsymbol{\theta}}(s) = 10/(1 + \exp(-\theta_1 \times (\boldsymbol{\psi}(s) - \theta_0))) + \theta_2$ where $\boldsymbol{\psi}(s)$ indicates the number of coins present in state $s$. (b) shows the NLL value of 50 expert demonstrations for $\{\theta_0, \theta_1\}$ with no translation while (c) shows the same for translation by a value of 2. (d) $\boldsymbol{\theta}^a$ and $\boldsymbol{\theta}^b$ are policy invariant and map to the same point in the projected space. $\boldsymbol{\theta}^c$ and $\boldsymbol{\theta}^d$ have a similar likelihood and are mapped to nearby positions.

reward functions in the form of a GP posterior. Uncertainty estimates of the NLL for each reward function enable downstream analysis and existing methods such as active learning [23] and active teaching [9] can be used to further narrow down these solutions. Given the benefits above, it may appear surprising that BO has not yet been applied to IRL, considering its application to many different domains [35]. A possible reason may be that BO does not work "out-of-the-box" for IRL despite its apparent suitability. Indeed, our initial naïve application of BO to IRL failed to produce good results.

Further investigation revealed that standard kernels were unsuitable for representing the covariance structure in the space of reward functions. In particular, they ignore policy invariance [3] where a reward function maintains its optimal policy under certain operations such as linear translation. Leveraging on this insight, we contribute a novel $\rho$-projection that remedies this problem. Briefly, the $\rho$-projection maps policy invariant reward functions to a single point in a new representation space where nearby points share similar NLL; Fig. 1 illustrates this key idea on a Gridworld environment.[1] With the $\rho$-projection in hand, standard stationary kernels (such as the popular RBF) can be applied in a straightforward manner. We provide theoretical support for this property and experiments on a variety of environments (both discrete and continuous, with model-based and model-free settings) show that our BO-IRL algorithm (with $\rho$-projection) efficiently captures the correlation structure of the reward space and outperforms representative state-of-the-art methods.

## 2 Preliminaries and Background

**Markov Decision Process (MDP).** An MDP is defined by a tuple $\mathcal{M} : \langle \mathcal{S}, \mathcal{A}, \mathcal{P}, R, \gamma \rangle$ where $\mathcal{S}$ is a finite set of states, $\mathcal{A}$ is a finite set of actions, $\mathcal{P}(s'|s, a)$ is the conditional probability of next state $s'$ given current state $s$ and action $a$, $R : \mathcal{S} \times \mathcal{A} \times \mathcal{S} \to \mathbb{R}$ denotes the reward function, and $\gamma \in (0, 1)$ is the discount factor. An optimal policy $\pi^*$ is a policy that maximizes the expected sum of discounted rewards $\mathbb{E}\left[\sum_{t=0}^{\infty} \gamma^t R(s_t, a_t, s_{t+1}) | \pi, \mathcal{M}\right]$. The task of finding an optimal policy is referred to as policy optimization. If the MDP is fully known, then policy optimization can be performed via dynamic programming. In model-free settings, RL algorithms such as proximal policy optimization [34] can be used to obtain a policy.

**Inverse Reinforcement Learning (IRL).** Often, it is difficult to manually specify or engineer a reward function. Instead, it may be beneficial to learn it from experts. The problem of inferring the unknown reward function from a set of (near) optimal demonstrations is known as IRL. The learner is

provided with an MDP without a reward function, $\mathcal{M} \setminus R$, and a set $\mathcal{T} \triangleq \{\tau_i\}_{i=1}^N$ of $N$ trajectories. Each trajectory $\tau \triangleq \{(s_t, a_t)\}_{t=0}^{L-1}$ is of length $L$.

Similar to prior work, we assume that the reward function can be represented by a real vector $\boldsymbol{\theta} \in \Theta \subseteq \mathbb{R}^d$ and is denoted by $R_{\boldsymbol{\theta}}(s, a, s')$. Overloading our notation, we denote the discounted reward of a trajectory $\tau$ as $R_{\boldsymbol{\theta}}(\tau) \triangleq \sum_{t=0}^{L-1} \gamma^t R_{\boldsymbol{\theta}}(s_t, a_t, s_{t+1})$. In the maximum entropy framework [38], the probability $p_{\boldsymbol{\theta}}(\tau)$ of a given trajectory is related to its discounted reward as follows:

$$p_{\boldsymbol{\theta}}(\tau) = \exp(R_{\boldsymbol{\theta}}(\tau))/Z(\boldsymbol{\theta}) \tag{1}$$

where $Z(\boldsymbol{\theta})$ is the partition function that is intractable in most practical scenarios. The optimal parameter $\boldsymbol{\theta}^*$ is given by $\operatorname{argmin}_{\boldsymbol{\theta}} L_{\text{IRL}}(\boldsymbol{\theta})$ where

$$L_{\text{IRL}}(\boldsymbol{\theta}) \triangleq -\sum_{\tau \in \mathcal{T}} \sum_{t=0}^{L-2} [\log(\pi_{\boldsymbol{\theta}}^*(s_t, a_t)) + \log(\mathcal{P}(s_{t+1}|s_t, a_t))] \tag{2}$$

is the negative log-likelihood (NLL) and $\pi_{\boldsymbol{\theta}}^*$ is the optimal policy computed using $R_{\boldsymbol{\theta}}$.

## 3  Bayesian Optimization-Inverse Reinforcement Learning (BO-IRL)

Recall that IRL algorithms take as input an MDP $\mathcal{M} \setminus R$, a space $\Theta$ of reward function parameters, and a set $\mathcal{T}$ of $N$ expert demonstrations. We follow the maximum entropy framework where the optimal parameter $\boldsymbol{\theta}^*$ is given by $\operatorname{argmin}_{\boldsymbol{\theta}} L_{\text{IRL}}(\boldsymbol{\theta})$ and $L_{\text{IRL}}(\boldsymbol{\theta})$ takes the form shown in (2). Unfortunately, calculating $\pi_{\boldsymbol{\theta}}^*$ in (2) is expensive, which renders exhaustive exploration of the reward function space infeasible. To mitigate this expense, we propose to leverage Bayesian optimization (BO) [26].

Bayesian optimization is a general sequential strategy for finding a global optimum of an expensive black-box function $f : \mathcal{X} \to \mathbb{R}$ defined on some bounded set $\mathcal{X} \in \mathbb{R}^d$. In each iteration $t = 1, \ldots, T$, an input query $\mathbf{x}_t \in \mathcal{X}$ is selected to evaluate the value of $f$ yielding a noisy output $y_t \triangleq f(\mathbf{x}_t) + \epsilon$ where $\epsilon \sim \mathcal{N}(0, \sigma^2)$ is i.i.d. Gaussian noise with variance $\sigma^2$. Since evaluation of $f$ is expensive, a surrogate model is used to strategically select input queries to approach the global minimizer $\mathbf{x}^* = \operatorname{argmin}_{\mathbf{x} \in \mathcal{X}} f(\mathbf{x})$. The candidate $\mathbf{x}_t$ is typically found by maximizing an *acquisition function*. In this work, we use a Gaussian process (GP) [36] as the surrogate model and expected improvement (EI) [26] as our acquisition function.

**Gaussian process (GP).**   A GP is a collection of random variables $\{f(\mathbf{x})\}_{\mathbf{x} \in \mathcal{X}}$ where every finite subset follows a multivariate Gaussian distribution. A GP is fully specified by its prior mean $\mu(\mathbf{x})$ and covariance $k(\mathbf{x}, \mathbf{x}')$ for all $\mathbf{x}, \mathbf{x}' \in \mathcal{X}$. In typical settings, $\mu(\mathbf{x})$ is often set to zero and the kernel function $k(\mathbf{x}, \mathbf{x}')$ is the primary ingredient. Given a column vector $\mathbf{y}_T \triangleq [y_t]_{t=1..T}^\top$ of noisy observations of $f$ at inputs $\mathbf{x}_1, \ldots, \mathbf{x}_T$ obtained after $T$ evaluations, a GP permits efficient computation of its posterior for any input $\mathbf{x}$. The GP posterior is a Gaussian with posterior mean and variance

$$\begin{aligned} \mu_T(\mathbf{x}) &\triangleq \mathbf{k}_T(\mathbf{x})^\top + (\mathbf{K}_T + \sigma^2 I)^{-1} \mathbf{y}_T \\ \sigma_T^2(\mathbf{x}) &\triangleq k(\mathbf{x}, \mathbf{x}) - \mathbf{k}_T(\mathbf{x})^\top (\mathbf{K}_T + \sigma^2 I)^{-1} \mathbf{k}_T(\mathbf{x}) \end{aligned} \tag{3}$$

where $\mathbf{K} \triangleq [k(\mathbf{x}_t, \mathbf{x}_{t'})]_{t,t'=1,\ldots,T}$ is the kernel matrix and $\mathbf{k}(\mathbf{x}) \triangleq [k(\mathbf{x}_t, \mathbf{x})]_{t=1,\ldots,T}^\top$ is the vector of cross-covariances between $\mathbf{x}$ and $\mathbf{x}_t$.

**Expected Improvement (EI).**   EI attempts to find a new candidate input $\mathbf{x}_t$ at iteration $t$ that maximizes the expected improvement over the best value seen thus far. Given the current GP posterior and $\mathbf{x}_{\text{best}} \triangleq \operatorname{argmax}_{\mathbf{x} \in \{\mathbf{x}_1, \ldots, \mathbf{x}_{t-1}\}} f(\mathbf{x})$, the next $\mathbf{x}_t$ is found by maximizing

$$a_{\text{EI}}(x) \triangleq \sigma_{t-1}(\mathbf{x})[\gamma_{t-1}(\mathbf{x}) \Phi(\gamma_{t-1}(\mathbf{x})) + \mathcal{N}(\gamma_{t-1}(\mathbf{x}); 0, 1)] \tag{4}$$

where $\Phi(x)$ is the cumulative distribution function of the standard Gaussian and $\gamma_t(\mathbf{x}) \triangleq (f(\mathbf{x}_{\text{best}} - \mu_t(\mathbf{x}))/\sigma_t(\mathbf{x})$ is a $Z$-score.

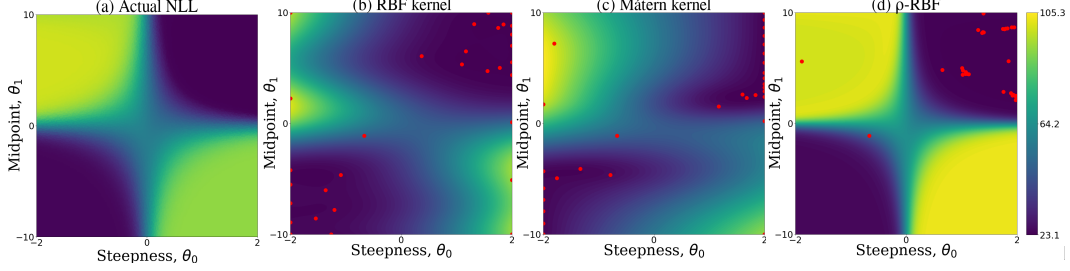

Figure 2: The NLL for the Gridworld problem across different reward parameters. (a) The true NLL. The GP posterior means obtained using the (b) RBF, (c) Matérn, and (d) $\rho$-RBF kernels with 30 iterations of BO-IRL.

**Specializing BO for IRL.** To apply BO to IRL, we set the function $f$ to be the IRL loss, i.e., $f(\boldsymbol{\theta}) = L_{\text{IRL}}(\boldsymbol{\theta})$, and specify the kernel function $k(\boldsymbol{\theta}, \boldsymbol{\theta}')$ in the GP. The latter is a crucial choice; since the kernel encodes the prior covariance structure across the reward parameter space, its specification can have a dramatic impact on search performance. Unfortunately, as we will demonstrate, popular stationary kernels are generally unsuitable for IRL. The remainder of this section details this issue and how we can remedy it via a specially-designed projection.

## 3.1 Limitations of Standard Stationary Kernels: An Illustrative Example

As a first attempt to optimize $L_{\text{IRL}}$ using BO, one may opt to parameterize the GP surrogate function with standard stationary kernels, which are functions of $\boldsymbol{\theta} - \boldsymbol{\theta}'$. For example, the radial basis function (RBF) kernel is given by

$$k_{\text{RBF}}(\boldsymbol{\theta}, \boldsymbol{\theta}') = \exp(-\|\boldsymbol{\theta} - \boldsymbol{\theta}'\|^2 / 2l^2) \tag{5}$$

where the lengthscale $l$ captures how far one can reliably extrapolate from a given data point. While simple and popular, the RBF is a poor choice for capturing covariance structure in the reward parameter space. To elaborate, the RBF kernel encodes the notion that reward parameters which are closer together (in terms of squared Euclidean distance) have similar $L_{\text{IRL}}$ values. However, this structure does not generally hold true in an IRL setting due to policy invariance; in our Gridworld example, $L_{\text{IRL}}(\boldsymbol{\theta}^a)$ is the same as $L_{\text{IRL}}(\boldsymbol{\theta}^b)$ despite $\boldsymbol{\theta}^a$ and $\boldsymbol{\theta}^b$ being far apart (see Fig. 1b). Indeed, Fig. 2b illustrates that applying BO with the RBF kernel yields a poor GP posterior approximation to the true NLLs. The same effect can be seen for the Matérn kernel in Fig. 2c.

## 3.2 Addressing Policy Invariance with the $\rho$-Projection

The key insight of this work is that better exploration can be achieved via an alternative representation of reward functions that mitigates policy invariance associated with IRL [3]. Specifically, we develop the $\rho$-projection whose key properties are that (a) policy invariant reward functions are mapped to a single point and (b) points that are close in its range correspond to reward functions with similar $L_{\text{IRL}}$. Effectively, the $\rho$-projection maps reward function parameters into a space where standard stationary kernels are able to capture the covariance between reward functions. For expositional simplicity, let us first consider the special case where we have only one expert demonstration.

**Definition 1** *Consider an MDP $\mathcal{M}$ with reward $R_{\boldsymbol{\theta}}$ and a single expert trajectory $\tau$. Let $\mathcal{F}(\tau)$ be a set of $M$ uniformly sampled trajectories from $\mathcal{M}$ with the same starting state and length as $\tau$. Define the $\rho$-projection $\rho_\tau : \Theta \to \mathbb{R}$ as*

$$\begin{aligned}
\rho_\tau(\boldsymbol{\theta}) &\triangleq \frac{p_{\boldsymbol{\theta}}(\tau)}{p_{\boldsymbol{\theta}}(\tau) + \sum_{\tau' \in \mathcal{F}(\tau)} p_{\boldsymbol{\theta}}(\tau')} \\
&= \frac{\exp(R_{\boldsymbol{\theta}}(\tau)/Z(\boldsymbol{\theta}))}{\exp(R_{\boldsymbol{\theta}}(\tau)/Z(\boldsymbol{\theta})) + \sum_{\tau' \in \mathcal{F}(\tau)} \exp(R_{\boldsymbol{\theta}}(\tau')/Z(\boldsymbol{\theta}))} \\
&= \frac{\exp(R_{\boldsymbol{\theta}}(\tau))}{\exp(R_{\boldsymbol{\theta}}(\tau)) + \sum_{\tau' \in \mathcal{F}(\tau)} \exp(R_{\boldsymbol{\theta}}(\tau'))} \,.
\end{aligned} \tag{6}$$

The first equality in (6) is a direct consequence of the assumption that the distribution of trajectories in MDP $\mathcal{M}$ follows (1) from the maximum entropy IRL framework. It can be seen from the second equality in (6) that an appealing property of $\rho$-projection is that the partition function is canceled off from the numerator and denominator, thereby eliminating the need to approximate it. Note that the $\rho$-projection is *not* an approximation of $p(\tau)$ despite the similar forms. $\mathcal{F}(\tau)$ in the denominator of $\rho$-projection is sampled to have the same starting point and length as $\tau$; as such, it may not cover the space of all trajectories and hence does not approximate $Z(\boldsymbol{\theta})$ even with large $M$. We will discuss below how the $\rho$-projection achieves the aforementioned properties. Policy invariance can occur due to multiple causes and we begin our discussion with a common class of policy invariant reward functions, namely, those resulting from potential-based reward shaping (PBRS) [28].

**$\rho$-Projection of PBRS-Based Policy Invariant Reward Functions.** Reward shaping is a method used to augment the reward function with additional information (referred to as a shaping function) without changing its optimal policy [24]. Designing a reward shaping function can be thought of as the inverse problem of identifying the underlying cause of policy invariance. Potential-based reward shaping (PBRS) [28] is a popular shaping function that provides theoretical guarantees for single-objective single-agent domains. We summarize the main theoretical result from [28] below:

**Theorem 1** *Consider an MDP $\mathcal{M}_0 : \langle S, A, T, \gamma, R_0 \rangle$. We define PBRS $F : S \times A \times S \to \mathbb{R}$ to be a function of the form $F(s, a, s') \triangleq \gamma\phi(s') - \phi(s)$ where $\phi(s)$ is any function of the form $\phi : S \to \mathbb{R}$. Then, for all $s, s' \in S$ and $a \in A$, the following transformation from $R_0$ to $R$ is sufficient to guarantee that every optimal policy in $\mathcal{M}_0$ is also optimal in MDP $\mathcal{M} : \langle S, A, T, \gamma, R \rangle$:*

$$R(s, a, s') \triangleq R_0(s, a, s') + F(s, a, s') = R_0(s, a, s') + \gamma\phi(s') - \phi(s) . \tag{7}$$

**Remark 1** The work of [28] has proven Theorem 1 for the special case of deterministic policies. However, this theoretical result also holds for stochastic policies, as shown in Appendix A.

**Corollary 1** *Given a reward function $R(s, a, s')$, any reward function $\hat{R}(s, a, s') \triangleq R(s, a, s) + c$ is policy invariant to $R(s, a, s')$ where $c$ is a constant. This is a special case of PBRS where $\phi(s)$ is a constant.*

The following theorem states that $\rho$-projection maps reward functions that are shaped using PBRS to a single point given sufficiently long trajectories:

**Theorem 2** *Let $R_{\boldsymbol{\theta}}$ and $R_{\hat{\boldsymbol{\theta}}}$ be reward functions that are policy invariant under the definition in Theorem 1. Then, w.l.o.g., for a given expert trajectory $\tau$ with length L,*

$$\lim_{L\to\infty} \rho_\tau(\hat{\boldsymbol{\theta}}) = \rho_\tau(\boldsymbol{\theta}) . \tag{8}$$

Its proof is in Appendix B. In brief, when summing up $F(s, a, s')$ (from Theorem 1) across the states and actions in a trajectory, most terms cancel out leaving only two terms: (a) $\phi(s_0)$ which depends on the start state $s_0$ and (b) $\gamma^L\phi(s_L)$ which depends on the end state $s_L$. With a sufficiently large $L$, the second term reaches zero. Our definition of $\rho_\tau(\boldsymbol{\theta})$ assumes that $s_0$ is the same for all trajectories. As a result, the influence of these two terms and by extension, the influence of the reward shaping function is removed by the $\rho$-projection.

**Corollary 2** *$\rho_\tau(\hat{\boldsymbol{\theta}}) = \rho_\tau(\boldsymbol{\theta})$ if (a) $R_{\boldsymbol{\theta}}$ and $R_{\hat{\boldsymbol{\theta}}}$ are only state dependent or (b) all $\tau' \in \mathcal{F}(\tau)$ have the same end state as $\tau$ in addition to the same starting state and same length.*

Its proof is in Appendix C.

**$\rho$-Projection of Other Classes of Policy Invariance.** There may exist other classes of policy invariant reward functions for a given IRL problem. How does the $\rho$-projection handle these policy invariant reward functions? We argue that $\rho$-projection indeed maps all policy invariant reward functions (regardless of their function class) to a single point if (1) holds true. Definition 1 casts the $\rho$-projection as a function of the likelihood of given (fixed) trajectories. Hence, the $\rho$-projection is identical for reward functions that are policy invariant since the likelihood of a fixed set of trajectories is the same for such reward functions. The $\rho$-projection can also be interpreted as a ranking function between the expert demonstrations and uniformly sampled trajectories, as shown in [8]. A high

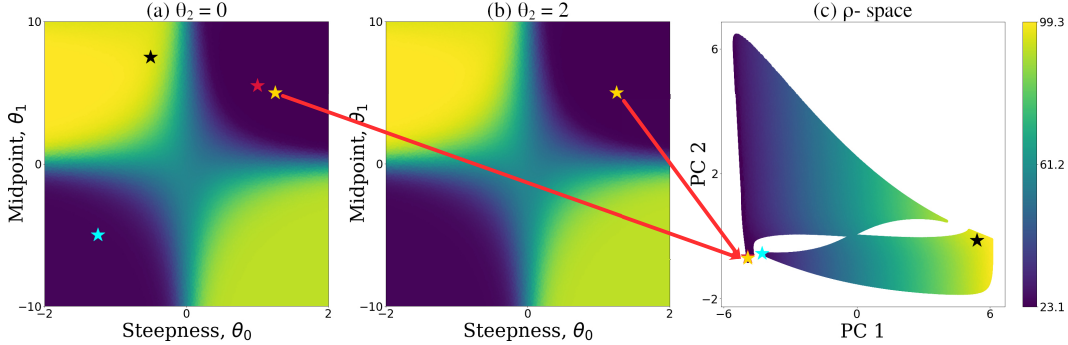

Figure 3: Capturing policy invariance. (a) and (b) represent $L_{\text{IRL}}$ values at two different $\theta_2$. (c) shows the corresponding $\rho$-space where the policy invariant $\boldsymbol{\theta}$ parameters are mapped to the same point.

$\rho$-projection implies a higher preference for expert trajectories over uniformly sampled trajectories with this relative preference decreasing with lower $\rho$-projection. This ensures that reward functions with similar likelihoods are mapped to nearby points.

### 3.3   $\rho$-RBF: Using the $\rho$-Projection in BO-IRL

For simplicity, we have restricted the above discussion to a single expert trajectory $\tau$. In practice, we typically have access to $K$ expert trajectories and can project $\boldsymbol{\theta}$ to a $K$-dimensional vector $[\rho_{\tau^k}(\boldsymbol{\theta})]_{k=1}^{K}$. The similarity of two reward functions can now be assessed by the Euclidean distance between their projected points. In this work, we use a simple RBF kernel after the $\rho$-projection, which results in the $\rho$-RBF kernel; other kernels can also be used. Algorithm 2 in Appendix E describes in detail the computations required by the $\rho$-RBF kernel. With the $\rho$-RBF kernel, BO-IRL follows standard BO practices with EI as an acquisition function (see Algorithm 1 in Appendix E). BO-IRL can be applied to both discrete and continuous environments, as well as model-based and model-free settings.

Fig. 3 illustrates the $\rho$-projection "in-action" using the Gridworld example. Recall the reward function in this environment is parameterized by $\boldsymbol{\theta} = \{\theta_0, \theta_1, \theta_2\}$. By varying $\theta_2$ (translation) while keeping $\{\theta_0, \theta_1\}$ constant, we generate reward functions that are policy invariant, as per Corollary 1. The yellow stars are two such policy invariant reward functions (with fixed $\{\theta_0, \theta_1\}$ and two different values of $\theta_2$) that share identical $L_{\text{IRL}}$ (i.e., indicated by color). Fig. 3c shows a PCA-reduced representation of the 20-dimensional $\rho$-space (i.e., the range of the $\rho$-projection). These two reward parameters are mapped to a single point. Furthermore, reward parameters that are similar in likelihood (red, blue, and yellow stars) are mapped close to one other. Using the $\rho$-RBF in BO yields a better posterior and samples, as illustrated in Fig. 2d.

### 3.4   Related Work

Our approach builds upon the methods and tools developed to address IRL, in particular, maximum entropy IRL (ME-IRL) [38]. However, compared to ME-IRL and its deep learning variant: maximum entropy deep IRL (deep ME-IRL) [37], our BO-based approach can reduce the number of (expensive) exact policy evaluations via better exploration. Newer approaches such as guided cost learning (GCL) [12] and adversarial IRL (AIRL) [14] avoid exact policy optimization by approximating the policy using a neural network that is learned along with the reward function. However, the quality of the solution obtained depends on the heuristics used and similar to ME-IRL: These methods return a single solution. In contrast, BO-IRL returns the best-seen reward function (possibly a set) along with the GP posterior which models $L_{\text{IRL}}$.

A related approach is Bayesian IRL (BIRL) [32] which incorporates prior information and returns a posterior over reward functions. However, BIRL attempts to obtain the entire posterior and utilizes a random policy walk, which is inefficient. In contrast, BO-IRL focuses on regions with high likelihood. GP-IRL [20] utilizes a GP as the reward function, while we use a GP as a surrogate for

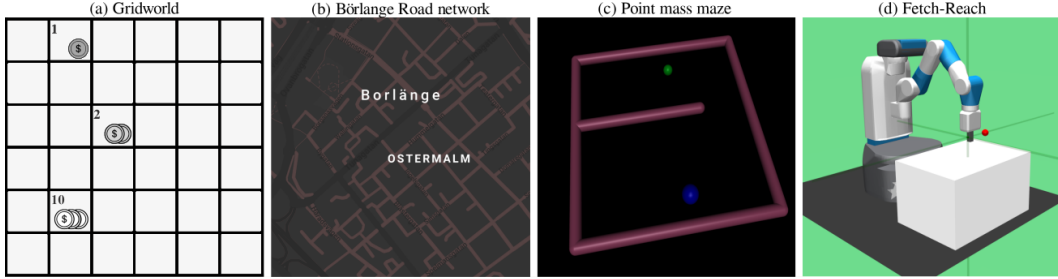

Figure 4: Environments used in our experiments. (a) Gridworld environment, (b) Börlange road network, (c) Point Mass Maze, and (d) Fetch-Reach task environment from OpenAI Gym.

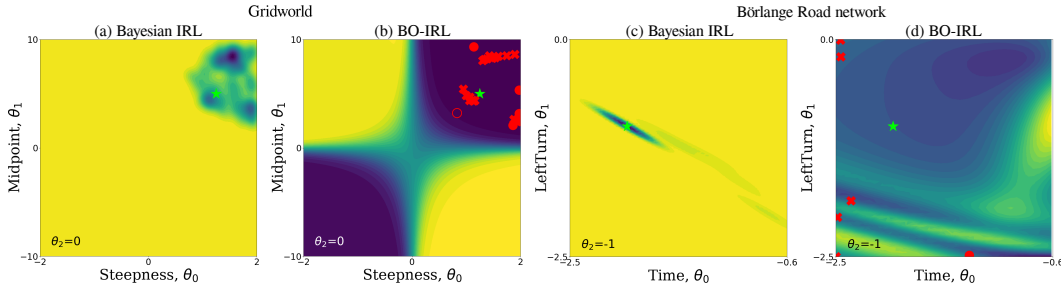

Figure 5: Posterior distribution over reward functions recovered by BIRL for (a) Gridworld environment and (c) Börlange road network, respectively. The GP posteriors over NLL learned by BO-IRL for the same environments are shown in (b) and (d). The red crosses represent samples selected by BO that have NLL better than the expert's true reward function. The red filled dots and red empty dots are samples whose NLL are similar to the expert's NLL, i.e., less than 1% and 10% larger, respectively. The green ⋆ indicates the expert's true reward function.

$L_{\text{IRL}}$. Compatible reward IRL (CR-IRL) [25] can also retrieve multiple reward functions that are consistent with the policy learned from the demonstrations using behavioral cloning. However, since demonstrations are rarely exhaustive, behavioral cloning can overfit, thus leading to an incorrect policy. Recent work has applied adversarial learning to derive policies, specifically, by generative adversarial imitation learning (GAIL) [16]. However, GAIL directly learns the expert's policy (rather the a reward function) and is not directly comparable to BO-IRL.

## 4 Experiments and Discussion

In this section, we report on experiments designed to answer two primary questions:

**Q1** Does BO-IRL with $\rho$-RBF uncover multiple reward functions consistent with the demonstrations?

**Q2** Is BO-IRL able to find good solutions compared to other IRL methods while reducing the number of policy optimizations required?

Due to space constraints, we focus on the key results obtained. Additional results and plots are available in Appendix F.

**Setup and Evaluation.** Our experiments were conducted using the four environments shown in Fig. 4: two model-based discrete environments, Gridworld and Börlange road network [13], and two model-free continuous environments, Point Mass Maze [14] and Fetch-Reach [31]. Evaluation for the Fetch-Reach task environment was performed by comparing the success rate of the optimal policy $\pi_{\hat{\theta}}$ obtained from the learned reward $\hat{\theta}$. For the other environments, we have computed the expected sum of rewards (ESOR) which is the average ground truth reward that an agent receives

while traversing a trajectory sampled using $\pi_{\hat{\theta}}$. For BO-IRL, the best-seen reward function is used for the ESOR calculation. More details about the experimental setup is available in Appendix D.

**BO-IRL Recovers Multiple Regions of High Likelihood.** To answer **Q1**, we examine the GP posteriors learned by BO-IRL (with $\rho$-RBF kernel) and compare them against Bayesian IRL (BIRL) with uniform prior [32]. BIRL learns a posterior distribution over reward functions, which can also be used to identify regions with high-probability reward functions. Figs. 5a and 5c show that BIRL assigns high probability to reward functions adjacent to the ground truth but ignores other equally probable regions. In contrast, BO-IRL has identified multiple regions of high likelihood, as shown in Figs. 5b and 5d. Interestingly, BO-IRL has managed to identify multiple reward functions with lower NLL than the expert's true reward (as shown by red crosses) in both environments. For instance, the linear "bands" of low NLL values at the bottom of Fig. 5d indicate that the travel patterns of the expert agent in the Börlange road network can be explained by any reward function that correctly trades off the time needed to traverse a road segment with the number of left turns encountered; left-turns incur additional time penalty due to traffic stops.

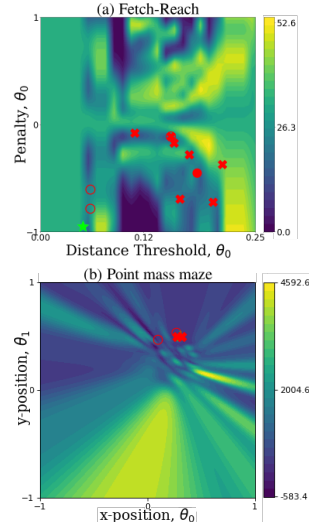

Figure 6: BO-IRL's GP posteriors for (a) Fetch-Reach task environment and (b) Point Mass Maze.

Figs. 6a and 6b show the GP posterior learned by BO-IRL for the two continuous environments. The Fetch-Reach task environment has a discontinuous reward function of the distance threshold and penalty. As seen in Fig. 6a, the reward function space in the Fetch-Reach task environment has multiple disjoint regions of high likelihood, hence making it difficult for traditional IRL algorithms to converge to the true solution. Similarly, multiple regions of high likelihood are also observed in the Point Mass Maze setting (Fig. 6b).

**BO-IRL Performs Well with Fewer Iterations Relative to Existing Methods.** In this section, we describe experimental results related to **Q2**, i.e., whether BO-IRL is able to find high-quality solutions within a given budget, as compared to other representative state-of-the-art approaches. We compare BO-IRL against BIRL, guided cost learning (GCL) [12] and adversarial IRL (AIRL) [14]. As explained in Appendix D.5, deep ME-IRL [37] has failed to give meaningful results across all the settings and is hence not reported. Note that GCL and AIRL do not use explicit policy evaluations and hence take less computation time. However, they only return a *single* reward function. As such, they are not directly comparable to BO-IRL, but serve to illustrate the quality of solutions obtained using recent approximate single-reward methods. BO-IRL with RBF and Matérn kernels do not have the overhead of calculating the projection function and therefore has a faster computation time. However, as seen from Fig. 2, these kernels fail to correctly characterize the reward function space correctly.

We ran BO-IRL with the RBF, Matérn, and $\rho$-RBF kernels. Table 1 summarizes the results for Gridworld environment, Börlange road network, and Point Mass Maze. Since no ground truth reward is available for the Börlange road network, we used the reward function in [13] and generated artificial trajectories.[2] BO-IRL with $\rho$-RBF reached expert's ESOR with fewer iterations than the other tested algorithms across all the settings. BIRL has a higher success rate in Gridworld environment compared to our method; however, it requires a significantly higher number of iterations with each iteration involving expensive exact policy optimization. It is also worth noting that AIRL and GCL are unable to exploit the transition dynamics of the Gridworld environment and Börlange road network settings. This in turn results in unnecessary querying of the environment for additional trajectories to approximate the policy function. BO-IRL is flexible to handle both model-free and model-based environments by an appropriate selection of the policy optimization method.

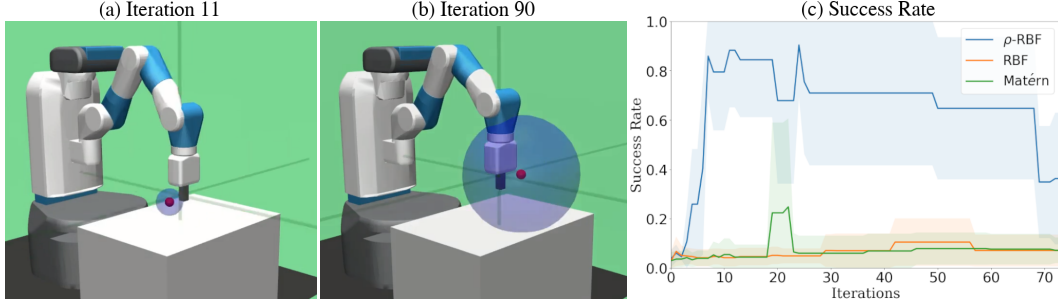

Figure 7: (a) and (b) indicate the learned distance threshold (blue sphere) for the Fetch-Reach task environment identified by BO-IRL at iterations 11 and 90, respectively. (c) shows the success rates evaluated using policies from the learned reward function. $\rho$-RBF kernel outperforms standard kernels.

Fig. 7c shows that policies obtained from rewards learned using $\rho$-RBF achieve higher success rates compared to other kernels in the Fetch-Reach task environment.[3] Interestingly, the success rate falls in later iterations due to the discovery of reward functions that are consistent with the demonstrations but do not align with the actual goal of the task. For instance, the NLL for Fig. 7b is less than that for Fig. 7a. However, the intention behind this task is clearly better captured by the reward function in Fig. 7a: The distance threshold from the target (blue circle) is small, hence indicating that the robot gripper has to approach the target. In comparison, the reward function in Fig. 7b encodes a large distance threshold, which rewards every action inside the blue circle. These experiments show that "blindly" optimizing NLL can lead to poor policies. The different solutions that are discovered by BO-IRL can be further analyzed downstream to select an appropriate reward function or to tweak state representations.

Table 1: Success rate (SR) and iterations required to achieve the expert's ESOR in Gridworld environment, Börlange road network, and Point Mass Maze. Best performance is in **bold**.

| Algorithm | Kernel | Gridworld | | Börlange | | Point mass maze | |
|---|---|---|---|---|---|---|---|
| | | SR | Iterations | SR | Iterations | SR | Iterations |
| | $\rho$-RBF | 70% | **16.0**±15.6 | **100%** | **2.0**±1.1 | **80%** | **51.4**±23.1 |
| BO-IRL | RBF | 50% | 30.0±34.4 | 80% | 9.5±6.3 | 20% | 28.0±4 |
| | Matérn | 60% | 22.2±12.2 | 100% | 5.6±3.8 | 20% | 56±29 |
| BIRL | | **80%** | 630.5±736.9 | 80% | 98±167.4 | | N.A. |
| AIRL | | 70% | 70.4±23.1 | 100% | 80±36.3 | 80% | 90.0±70.4 |
| GCL | | 40% | 277.5±113.1 | 80% | 375±68.7 | 0% | — |

## 5    Conclusion and Future Work

This paper describes a Bayesian Optimization approach to reward function learning called BO-IRL. At the heart of BO-IRL is our $\rho$-projection (and the associated $\rho$-RBF kernel) that enables efficient exploration of the reward function space by explicitly accounting for policy invariance. Experimental results are promising: BO-IRL uncovers multiple reward functions that are consistent with the expert demonstrations while reducing the number of exact policy optimizations. Moving forward, BO-IRL opens up new research avenues for IRL. For example, we plan to extend BO-IRL to handle higher-dimensional reward function spaces, batch modes, federated learning and nonmyopic settings where recently developed techniques (e.g., [10, 11, 17, 18, 21, 33]) may be applied.

## Broader Impact

It is important that our autonomous agents operate with the correct objectives to ensure that they exhibit appropriate and trustworthy behavior (ethically, legally, etc.) [19]. This issue is gaining broader significance as autonomous agents are increasingly deployed in real-world settings, e.g., in the form of autonomous vehicles, intelligent assistants for medical diagnosis, and automated traders.

However, specifying objectives is difficult, and as this paper motivates, reward function learning via demonstration likelihood optimization may also lead to inappropriate behavior. For example, our experiments with the Fetch-Reach environment shows that apparently "good" solutions in terms of NLL correspond to poor policies. BO-IRL takes one step towards addressing this issue by providing an efficient algorithm for returning more information about *potential* reward functions in the form of discovered samples and the GP posterior. This approach can help users further iterate to arrive at appropriate reward function, e.g., to avoid policies that cause expected or undesirable behavior.

As with other learning methods, there is a risk for misuse. This work does not consider constraints that limit the reward functions that can be learned. As such, users may teach the robots to perform unethical or illegal actions; consider the recent incident where users taught the Microsoft's chatbot Tay to spout racist and anti-social tweets. With robots that are capable of physical actions, consequences may be more severe, e.g., bad actors may teach the robot to cause both psychological and physical harm. A more subtle problem is that harmful policies may result *unintentionally* from misuse of BO-IRL, e.g., when the assumptions of the method do not hold. These issues point to potential future work on verification or techniques to enforce constraints in BO-IRL and other IRL algorithms.

## Acknowledgments and Disclosure of Funding

This research/project is supported by the National Research Foundation, Prime Minister's Office, Singapore under its Campus for Research Excellence and Technological Enterprise (CREATE) program, Singapore-MIT Alliance for Research and Technology (SMART) Future Urban Mobility (FM) IRG and the National Research Foundation, Singapore under its AI Singapore Programme (AISG Award No: AISG-RP-2019-011). Any opinions, findings and conclusions or recommendations expressed in this material are those of the author(s) and do not reflect the views of National Research Foundation, Singapore.

## Footnotes

[1]This Gridworld environment will be our running example throughout this paper.

[2]BO-IRL was also tested on the real-world trajectories from the Börlange road network dataset; see Fig. 11 in Appendix F.4.

[3] AIRL and GCL were not tested on the Fetch-Reach task environment as the available code was incompatible with the environment.

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
