[Supplementary Material · BO-IRL_Appendix.pdf]

# A   Proof of Remark 1

The work of [28] proves Theorem 1 for the special case of deterministic policies. However, it also holds for stochastic policies, as shown below.

Consider an MDP $\mathcal{M}_0 : \langle S, A, T, R_0, \gamma \rangle$ with its Q-function given by $Q_0$. Assuming a stochastic policy is considered, we can replace the max operator in the standard Bellman update for the Q-function with a Boltzmann operator, as shown below:

$$Q_0(s, a) \triangleq \mathbb{E}_{s' \sim T} \left[ R_0(s, a, s') + \gamma \sum_{a' \in A} \pi(s', a') \, Q_0(s', a') \right]$$

where

$$\pi(s', a') = \frac{\exp(Q_0(s', a'))}{\sum_{a'' \in A} \exp(Q_0(s', a''))} \ . \tag{9}$$

Subtracting a real-valued function $\phi(s)$ from $Q_0(s, a)$,

$$
\begin{aligned}
& Q_0(s, a) - \phi(s) \\
&= \mathbb{E}_{s' \sim T} \left[ R_0(s, a, s') - \phi(s) + \gamma \sum_{a' \in A} \pi(s', a') \, Q_0(s', a') \right] \\
&= \mathbb{E}_{s' \sim T} \left[ R_0(s, a, s') - \phi(s) + \gamma \phi(s') - \gamma \phi(s') + \gamma \sum_{a' \in A} \pi(s', a') \, Q_0(s', a') \right] \\
&= \mathbb{E}_{s' \sim T} \left[ R_0(s, a, s') - \phi(s) + \gamma \phi(s') + \gamma \sum_{a' \in A} \pi(s', a') \, (Q_0(s', a') - \phi(s')) \right] .
\end{aligned}
\tag{10}
$$

So, (9) can be rewritten as

$$\pi(s', a') = \frac{\exp(Q_0(s', a') - \phi(s'))}{\sum_{a'' \in A} \exp(Q_0(s', a'') - \phi(s'))} \ . \tag{11}$$

Let us define $Q(s, a) \triangleq Q_0(s, a) - \phi(s)$. Then,

$$\pi(s', a') = \frac{\exp(Q(s', a'))}{\sum_{a'' \in A} \exp(Q(s', a''))} \ . \tag{12}$$

Let us also define $R(s, a, s') \triangleq R_0(s, a, s') + \gamma \phi(s') - \phi(s')$. Substituting this definition along with (12) into (10),

$$Q(s, a) = \mathbb{E}_{s' \sim T} \left[ R(s, a, s') + \gamma \sum_{a' \in A} \pi(s', a') \, Q(s', a') \right]$$

which is the Bellman update (with Boltzmann operator) for MDP $\mathcal{M}_1 : \langle S, A, T, R, \gamma \rangle$. Therefore, by shaping $R(s, a, s') = R_0(s, a, s') + \gamma \phi(s') - \phi(s)$, the policy remains unchanged and the Q-value is modified to $Q_0(s, a) - \phi(s)$.

# B   Proof of Theorem 2

Consider a single trajectory $\tau$ with length $L$ sampled from $\mathcal{M}$. Let $\mathcal{F}(\tau)$ be any function used to generate $M \geq 1$ additional trajectories with the same starting state and length as $\tau$. $R_{\boldsymbol{\theta}}$ and $R_{\hat{\boldsymbol{\theta}}}$ are assumed to be policy invariant under Theorem 1. Then, without loss of generality, $R_{\hat{\boldsymbol{\theta}}}(s, a, s') = R_{\boldsymbol{\theta}}(s, a, s') + F(s, a, s')$ where $F$ takes the form specified in (7). Recall from Section 2 that

$$
\begin{aligned}
R_{\hat{\boldsymbol{\theta}}}(\tau) &= \sum_{t=0}^{L-1} \gamma^t R_{\hat{\boldsymbol{\theta}}}(s_t, a_t, s_{t+1}) \\
&= \sum_{t=0}^{L-1} \gamma^t \left( R_{\boldsymbol{\theta}}(s_t, a_t, s_{t+1}) + F(s_t, a_t, s_{t+1}) \right) \\
&= R_{\boldsymbol{\theta}}(\tau) + \sum_{t=0}^{L-1} \gamma^t F(s_t, a_t, s_{t+1}) \ .
\end{aligned}
\tag{13}
$$

Using the definition of $F(s, a, s')$ from Theorem 1,

$$\sum_{t=0}^{L-1} \gamma^t F(s_t, a_t, s_{t+1}) = \sum_{t=0}^{L-1} \gamma^t \left(\gamma \phi(s_{t+1}) - \phi(s_t)\right) = \gamma^L \phi(s_L) - \phi(s_0) \,. \tag{14}$$

Substituting (14) into (13),

$$R_{\hat{\boldsymbol{\theta}}}(\tau) = R_{\boldsymbol{\theta}}(\tau) + \gamma^L \phi(s_L) - \phi(s_0) \,.$$

Using the same reasoning, for all $\tau' \in \mathcal{F}(\tau)$,

$$R_{\hat{\boldsymbol{\theta}}}(\tau') = R_{\boldsymbol{\theta}}(\tau') + \gamma^L \phi(s'_L) - \phi(s_0) \,.$$

By definition of $\rho$-projection (Definition 1), $s_0$ is the same for $\tau$ and all $\tau' \in \mathcal{F}(\tau)$. Note that $s_L$ is the last state in trajectory $\tau$ and $s'_L$ is the last state in trajectory $\tau'$. Define $h \triangleq \gamma^L \phi(s_L) - \phi(s_0)$ and $k' \triangleq \phi(s'_L) - \phi(s_L)$. Then,

$$\begin{aligned} R_{\hat{\boldsymbol{\theta}}}(\tau) &= R_{\boldsymbol{\theta}}(\tau) + h \\ R_{\hat{\boldsymbol{\theta}}}(\tau') &= R_{\boldsymbol{\theta}}(\tau') + h + \gamma^L k' \,. \end{aligned} \tag{15}$$

Using the definition of $\rho$-projection (Definition 1) and (15),

$$\begin{aligned} \rho_\tau(\hat{\boldsymbol{\theta}}) &= \frac{\exp(R_{\hat{\boldsymbol{\theta}}}(\tau))}{\exp(R_{\hat{\boldsymbol{\theta}}}(\tau)) + \sum_{\tau' \in \mathcal{F}(\tau)} \exp(R_{\hat{\boldsymbol{\theta}}}(\tau'))} \\ &= \frac{\exp(R_{\boldsymbol{\theta}}(\tau) + h)}{\exp(R_{\boldsymbol{\theta}}(\tau) + h) + \sum_{\tau' \in \mathcal{F}(\tau)} \exp(R_{\boldsymbol{\theta}}(\tau') + h + \gamma^L k')} \\ &= \frac{\exp(R_{\boldsymbol{\theta}}(\tau))}{\exp(R_{\boldsymbol{\theta}}(\tau)) + \sum_{\tau' \in \mathcal{F}(\tau)} \exp(R_{\boldsymbol{\theta}}(\tau') + \gamma^L k')} \,. \end{aligned} \tag{16}$$

Since $0 \le \gamma < 1$, $\gamma^L \to 0$ as $L \to \infty$. Therefore,

$$\begin{aligned} \lim_{L \to \infty} \rho_\tau(\hat{\boldsymbol{\theta}}) &= \frac{\exp(R_{\boldsymbol{\theta}}(\tau))}{\exp(R_{\boldsymbol{\theta}}(\tau)) + \sum_{\tau' \in \mathcal{F}(\tau)} \exp(R_{\boldsymbol{\theta}}(\tau'))} \\ &= \rho_\tau(\boldsymbol{\theta}) \,. \end{aligned}$$

## C   Proof of Corollary 2

Corollary 2 is a natural extension of Theorem 2. Let us consider the two cases separately.

**Case 1: Rewards only depend on the states.**   Using Corollary 1, the potential-based function $F(s, a, s')$ is a constant $c$. From (13),

$$\begin{aligned} R_{\hat{\boldsymbol{\theta}}}(\tau) &= \sum_{t=0}^{L-1} \gamma^t R_{\hat{\boldsymbol{\theta}}}(s_t, a_t, s_{t+1}) \\ &= R_{\boldsymbol{\theta}}(\tau) + \sum_{t=0}^{L-1} \gamma^t c \,. \end{aligned}$$

Similarly, for all $\tau' \in \mathcal{F}(\tau)$ with the same starting state $s_0$ and length $L$ as $\tau$,

$$R_{\hat{\boldsymbol{\theta}}}(\tau') = R_{\boldsymbol{\theta}}(\tau') + \sum_{t=0}^{L-1} \gamma^t c \,.$$

Let $c' \triangleq \sum_{t=0}^{L-1} \gamma^t c$. Using the definition of $\rho$-projection (Definition 1),

$$\begin{aligned} \rho_\tau(\hat{\boldsymbol{\theta}}) &= \frac{\exp(R_{\hat{\boldsymbol{\theta}}}(\tau))}{\exp(R_{\hat{\boldsymbol{\theta}}}(\tau)) + \sum_{\tau' \in \mathcal{F}(\tau)} \exp(R_{\hat{\boldsymbol{\theta}}}(\tau'))} \\ &= \frac{\exp(R_{\boldsymbol{\theta}}(\tau) + c')}{\exp(R_{\boldsymbol{\theta}}(\tau) + c') + \sum_{\tau' \in \mathcal{F}(\tau)} \exp(R_{\boldsymbol{\theta}}(\tau') + c')} \\ &= \frac{\exp(R_{\boldsymbol{\theta}}(\tau))}{\exp(R_{\boldsymbol{\theta}}(\tau)) + \sum_{\tau' \in \mathcal{F}(\tau)} \exp(R_{\boldsymbol{\theta}}(\tau'))} \\ &= \rho_\tau(\boldsymbol{\theta}) \,. \end{aligned}$$

**Case 2: All $\tau' \in \mathcal{F}(\tau)$ have the same end state $s_L$, starting state $s_0$, and length $L$ as $\tau$.** Recall that $k' = \phi(s'_L) - \phi(s_L)$ in (15): Since the end states are the same for $\tau$ and all $\tau' \in \mathcal{F}(\tau)$, $k' = 0$. Using this in (16),

$$
\begin{aligned}
\rho_\tau(\hat{\boldsymbol{\theta}}) &= \frac{\exp(R_{\hat{\boldsymbol{\theta}}}(\tau))}{\exp(R_{\hat{\boldsymbol{\theta}}}(\tau)) + \sum_{\tau' \in \mathcal{F}(\tau)} \exp(R_{\hat{\boldsymbol{\theta}}}(\tau'))} \\
&= \frac{\exp(R_{\boldsymbol{\theta}}(\tau) + h)}{\exp(R_{\boldsymbol{\theta}}(\tau) + h) + \sum_{\tau' \in \mathcal{F}(\tau)} \exp(R_{\boldsymbol{\theta}}(\tau') + h)} \\
&= \frac{\exp(R_{\boldsymbol{\theta}}(\tau))}{\exp(R_{\boldsymbol{\theta}}(\tau)) + \sum_{\tau' \in \mathcal{F}(\tau)} \exp(R_{\boldsymbol{\theta}}(\tau'))} \\
&= \rho_\tau(\boldsymbol{\theta}) \ .
\end{aligned}
$$

# D Experimental Setups

Figure 4 shows all the environments used in this study. We will now elaborate on the tasks, reward functions, and other details associated with each of these environments.

## D.1 Gridworld Environment

The Gridworld environment introduced in Fig. 1a is a synthetic experimental setup designed to show the similarities that exist in the reward function space $\Theta$. In this setup, each state $s$ is represented by a state feature $\phi(s)$ which corresponds to the number of gold coins in that state. We used a translated logistic function $R_{\boldsymbol{\theta}}(s) = 10/(1 + \exp(-\theta_1 \times (\phi(s) - \theta_0))) + \theta_2$ as reward function where $\theta_0$ controls the steepness of the logistic function, $\theta_1$ controls the midpoint, and $\theta_2$ translates the reward function. The ground truth values of these parameters are [1.25, 5.0, 0].

During the IRL training, a total of 50 expert trajectories of length 15 were used. For BO-IRL, a subset of randomly selected $K = 10$ trajectories were used for the calculation of $\rho$-projection at each trial. For each of these trajectories, $M = 5$ artificial trajectories of the same length and starting state were generated using a random policy walk. For BO initialization, points were selected from regions of high NLL to make the training challenging. BO optimizations ran for a budget of 100 evaluations while AIRL [14] and GCL [12] ran for 1000 iterations. Both the expert trajectories and initialization remained unchanged across the various tested algorithms for fair comparison. The bounds of $\Theta$ were set to

- steepness: $\theta_0 \in [-2, 2]$,
- midpoint: $\theta_1 \in [-10, 10]$, and
- translation: $\theta_2 \in [-4, 4]$.

## D.2 Börlange Road Network Dataset

The Börlange road network dataset contains road link information from the town of Börlange, Sweden. It contains 7288 links such that each link shares a vertex with at most 5 other links. A dummy link is also added from the given destination to indicate end of the trip, hence making the total number of links 7289. Features associated with traveling from a link $a$ to an adjacent link $b$ are available in the dataset.

We have modified this dataset to form an MDP where each state $s(a, b)$ corresponds to being in a particular link $b$ after traveling from an adjacent link $a$. Each $s(a, b)$ is defined by 4 state features:

1. Time to traverse $b$,
2. Is the turn from $a$ to $b$ a right-turn? Binary value with 0:yes and 1:no,
3. Constant 0 if $b$ is a sink state and 1 otherwise, and
4. Is the turn from $a$ to $b$ a u-turn? Binary value with 0:yes and 1:no.

The reward function is assumed to be a linear combination of these 4 state features with parameters $\theta_0, \theta_1, \theta_2, \theta_3$ corresponding to the features mentioned above in that order. $\theta_2$ allows us to penalize

any trips that contain too many road link traversals. In our experiments, $\theta_3$ was set to $-20$ and is not learned.

Furthermore, our action space contains 6 actions. Actions 0-5 at state $s(a, b)$ correspond to moving from $b$ to one of its adjacent links, $c$. In terms of transition probabilities, this corresponds to a deterministic transition from $s(a, b)$ to $s(b, c)$. Action 0 corresponds to moving to the adjacent link with the rightmost turn, followed by action 1 for the next right link, and so on. If the number of outgoing links of $b$ is less than 5, then it is assumed that the agent transitions back to state $s(a, b)$. Action 6 can be thought of as the "parking" action and is only valid for state $s(a, b)$ where $b$ is the dummy link. Taking action 6 in other states leads to a transition back to the same state. In total, this environment contains 20,199 states and 6 actions, hence making it challenging for exact policy optimization methods.

### D.2.1 Virtual Börlange Road Network Dataset

To test the quality of the reward function retrieved by BO-IRL, we need to calculate the expected sum of rewards (ESOR) and compare it to that of the expert. To do so, we need to have access to the ground truth reward function. Unfortunately, this is not available in the Börlange road network dataset. So, a simulation of the road network was constructed with the exact set of road links and connections. An artificial reward function with parameters $\theta_0 = -2, \theta_1 = -1, \theta_2 = -1, \theta_3 = -20$ was used and a new set of 20,000 expert trajectories were generated, which was further reduced to 635 informative trajectories. For BO-IRL, a subset of $K = 10$ expert trajectories were used for $\rho$-projection. For each expert trajectory, $M = 2000$ artificial trajectories were generated using a random policy. BO was initialized with points from regions of high NLL and executed for a budget of 50 evaluations. The following bounds of $\Theta$ were used:

- traverse time: $\theta_0 \in [-2.5, 2.5]$,
- right-turn: $\theta_1 \in [-2.5, 2.5]$,
- penalty: $\theta_2 \in [-2.5, 2.5]$, and
- u-turn: $\theta_3 = -20$.

### D.2.2 Real-World Börlange Road Network Dataset

The experiments from the virtual setting were repeated on the real-world trajectories available in the Börlange road network dataset. Only the negative log likelihood (NLL) was evaluated to verify whether BO-IRL converges faster than existing methods to an optimum. All the details for this setup was kept the same as the virtual setup, except for the number of expert trajectories. For expert trajectories, we selected 54 trajectories that end at a specific destination, but with different starting points.

### D.3 Fetch Robot Simulation

In this work, we utilize the Fetch Robot simulation which is a part of the OpenAI Gym [7]. In particular, we use the Fetch-Reach task environment. The goal of this task is to move the gripper of the Fetch robot to a goal position which is randomly populated in the 3D space at each iteration. The reward function is given by

$$R(s) \triangleq \begin{cases} 0 & \text{if } d(s) \leq \theta_0 , \\ \theta_1 & \text{otherwise} ; \end{cases}$$

where $d(s)$ corresponds to the distance between the gripper and the target at the given state $s$ and $\theta_0$ is a distance threshold beyond which a penalty value of $\theta_1$ is applied. The following bounds of $\Theta$ were used:

- threshold: $\theta_0 \in [0, 0.25]$, and
- penalty: $\theta_1 \in [-1.5, 1.5]$.

Since this is a model-free environment, we use proximal policy optimization (PPO) [34] to perform policy optimization. Due to the randomness inherent in PPO, we perform policy optimization 3 times and average the likelihood value when evaluating each reward function.

## D.4 Point Mass Maze

This environment closely follows the experimental setup in [14]. We have simplified the reward function from a deep neural network used in [14] to just the x-y position of the target location given by $\boldsymbol{\theta} = \{\theta_0, \theta_1\}$. As shown in Fig. 4c, the goal is to move the blue ball to the green target location. A state feature corresponding to state $s$ represents the current x-y location of the blue ball represented by $\tilde{\boldsymbol{\theta}}_s$. The reward function of a state $s$ is given by $R(s) \triangleq ||\boldsymbol{\theta} - \tilde{\boldsymbol{\theta}}_s||$. We use proximal policy optimization (PPO) [34] to perform policy optimization. The following bounds of $\Theta$ were used:

- threshold: $\theta_0 \in [-1, 1]$, and
- penalty: $\theta_1 \in [-1, 1]$.

## D.5 Maximum Entropy Deep IRL

We tested deep maximum entropy IRL (deep ME-IRL) [37] in the discrete environments, namely, the Gridworld environment and Börlange road network. In the Gridworld environment setting, it failed to reach the expert's ESOR across multiple trials. In the Börlange road network, the large state space made calculating the state-visitation frequency intractable. Deep ME-IRL is not compatible with continuous environments and was therefore not evaluated in the Point Mass Maze and Fetch-Reach task environment. Hence, it is omitted from Table 1.

# E  BO-IRL Algorithm

The full algorithm of BO-IRL with $\rho$-RBF kernel can be found in Algorithm 1. The algorithm can be split into four main phases. Phase 1 described in Algorithm 2 shows the steps involved in generating the $\boldsymbol{Z}$ dataset which contains $[(\tau^k, \mathcal{F}(\tau^k))]_{k=1}^{K}$ where $\tau^k$ is an expert trajectory. As mentioned in Definition 1, $\mathcal{F}(\tau^k)$ corresponds to $M$ sampled trajectories using an uniform policy with the same starting state and length as $\tau^k$.

In Phase 2, we define the two components of Bayesian Optimization, namely, the acquisition function and surrogate function. In our work, EI is used as the acquisition function. A GP with $\rho$-RBF kernel generated using the $\boldsymbol{Z}$ matrix from the previous phase is used as the surrogate function. For details on how to create this kernel, refer to Section 3.3.

Phases 3 and 4 follow the standard Bayesian Optimization practices. These involve initialization and optimization. During initialization, $n_{init}$ samples are drawn from the reward function space $\Theta$ to initialize the BO by updating the prior. In our experiments, we have collected a set of initialization points corresponding to high NLL values to make the training more challenging. During the optimization, the acquisition function is used to select the next reward function parameter to evaluate. After every evaluation, the GP posterior mean and standard deviation are updated using Bayes rule. You can find more information about standard BO practices from [6, 22, 26, 30].

# F  Additional Experimental Results

This section presents additional results obtained by running BO-IRL and other state-of-the-art IRL algorithms on the four environments shown in Fig. 4.

## F.1  GP Posterior Mean and Standard Deviation

Fig. 8 shows the posterior mean and standard deviation obtained using BO-IRL with $\rho$-RBF kernel for all the environments. As the plots show, the uncertainty in regions of high likelihood (low NLL) is low which indicates that BO has focused on uncovering regions of high likelihood. The top and bottom rows of Fig. 9 show the posterior mean obtained using BO-IRL with RBF and Matérn kernels. Comparing with the GP posterior mean obtained using $\rho$-RBF, we observe that RBF and Matérn need to explore the reward function space more exhaustively to identify multiple regions of high likelihood. Furthermore, the true likelihood values for the Gridworld environment setting (shown in Fig. 1b) matches closely with the posterior from $\rho$-RBF (Fig. 8a) when compared with that from RBF and Matérn (Fig. 9a). Finally, the standard kernels have also failed to capture a good reward function for the Point Mass Maze environment.

---

**Algorithm 1** BO-IRL

---

**Input:** expert demonstrations: $D$, budget: $B$, sizes: $K$, $M$, and $n_{init}$
$E \leftarrow \emptyset$ (to track all $\boldsymbol{\theta}$ values evaluated by BO)

{*Phase 1: Generate $\boldsymbol{Z}$*}
$\boldsymbol{Z} \leftarrow$ generateZ$(D, K, M)$ using Algorithm 2

{*Phase 2: Setup BO*}
BO Surrogate Function $\leftarrow$ GP with $\rho$-RBF kernel evaluated using $\boldsymbol{Z}$
BO Acquisition Function $\leftarrow$ Expected Improvement

{*Phase 3: Initialization*}
**repeat**
    Randomly select a reward-parameter $\boldsymbol{\theta}$
    Calculate optimal policy $\pi_{\boldsymbol{\theta}}$ using policy iteration (or policy gradient methods)
    Calculate NLL $\ell$ of $D$ using $\pi_{\boldsymbol{\theta}}$
    $E \leftarrow (\boldsymbol{\theta}, \ell)$
**until** Size of $E < n_{init}$
Update the GP Posterior using $E$

{*Phase 4: Optimization*}
**repeat**
    Using BO acquisition function, select next $\boldsymbol{\theta}$
    Calculate optimal policy $\pi_{\boldsymbol{\theta}}$ using policy iteration (or policy gradient methods)
    Calculate NLL $\ell$ of $D$ using $\pi_{\boldsymbol{\theta}}$
    $E \leftarrow (\boldsymbol{\theta}, \ell)$
    Update the GP Posterior using $E$
**until** Size of $E < B + n_{init}$

---

---

**Algorithm 2** generateZ

---

**Input:** expert demonstrations $D$, sizes: $K$ and $M$
$\boldsymbol{Z} \leftarrow \emptyset$
**for** $k = 1$ **to** $K$ **do**
    Randomly select a trajectory $\tau^k$ from $D$ without replacement
    $s \leftarrow$ Starting state of $\tau^k$
    $L \leftarrow$ Length of $\tau^k$
    $\mathcal{F}(\tau^k) \leftarrow \emptyset$
    **for** $i = 1$ **to** $M$ **do**
        Generate trajectory $\tau_i'^k$ with starting state $s$ and length $L$ by rolling out a uniform policy
        $\mathcal{F}(\tau^k) \leftarrow \mathcal{F}(\tau^k) \cup \tau_i'^k$
    **end for**
    $\boldsymbol{Z} \leftarrow \boldsymbol{Z} \cup (\tau^k, \mathcal{F}(\tau^k))$
**end for**
**Return** $\boldsymbol{Z}$

---

Figure 8: The GP posterior mean (top row) and standard deviation (bottom row) obtained after running BO-IRL with $\rho$-RBF kernel for all the tested environments. The red crosses represent samples selected by BO that have NLL better than the expert's true reward function. The red filled dots and red empty dots are samples whose NLL are similar to the expert's NLL, i.e., less than 1% and 10% larger, respectively. The green $\star$ indicates the expert's true reward function.

Figure 9: The posterior mean learned by BO-IRL with RBF (top row) and Matérn (bottom row) kernels for all the tested environments.

Figure 10: (a) Euclidean distance of the best reward function thus far from the ground truth target position. (b) ESOR value of the best reward function.

## F.2  Fetch-Reach Training Progress

A video file showing the best reward function obtained at each iteration is included along with the supplementary material. Recall that the reward function is parameterized by the distance threshold around the target location and the penalty associated with being outside the distance threshold. As shown in Figs. 7a and 7b, the blue circle is a visual representation of the distance threshold. The penalty values are reported at each frame of the video.

## F.3  Point Mass Maze

As reported in Table 1, Matérn outperforms $\rho$-RBF kernel for the Point Mass Maze environment. We believe this is due to $\rho$-RBF's ability to capture the correlation between NLL values better than Matern. Fig. 10a shows the Euclidean distance of the best reward function observed so far in the training (in terms of NLL) from the ground truth target position. Despite coming close to ground truth target position at iteration 4, BO-IRL with $\rho$-RBF kernel explores other regions of reward function space that have reward functions with lower NLL (iterations 9-20). However, the lower NLL values at the reward functions that are farther away from the ground truth do not translate directly into better ESOR, as can be seen in Fig. 10b.

## F.4  Börlange Road Network

Börlange road network dataset does not contain a ground truth reward function that generated the real-world data. Therefore, we created a simulated environment that mimics the road network in this dataset. Since no ground truth reward is available, we used the reward function in [13] and generated artificial trajectories. Table 1 shows the number of iterations required by the various algorithms to match the expert's performance. As observed, BO-IRL with $\rho$-RBF kernel outperforms the other methods.

With the new insight that BO-IRL matches the performance of the expert in the simulated Börlange road network, we tested BO-IRL against the real-world data. Performance was evaluated using the negative log likelihood (2) across iterations. Fig. 11 shows the performance of AIRL, GCL, and BO-IRL on the real-world data. GCL and AIRL converge slowly while BO-IRL finds points with low NLL within a few iterations. Amongst the BO-IRL kernels, $\rho$-RBF does not achieve the lowest NLL, but has comparable values to other kernels.

Figure 11: Negative log-likelihood of Börlange road network dataset at rewards retrieved by BO-IRL compared against that from AIRL and GCL. BO-IRL is able to converge to an optimal reward function faster than GCL or AIRL. Performance of $\rho$-RBF kernel was observed to be slightly worse than the other kernels in terms of the NLL values. AIRL eventually overfits and the training became unstable after 55 iterations.