[Reviews · NeurIPS 2020]

Review 1

Summary and Contributions: Thank you for your detailed rebuttal. The comparison to BIRL is appreciated. I'm still unsure if the difference in NLL is due to different likelihoods (boltzman vs. max ent), but I think this is a strong submission and a novel approach. --- This paper proposes a Bayesian optimization approach for learning a variety of reward functions that are likely given demonstrations. Bayesian optimization allows efficient exploration of the reward function space which is important since IRL methods are typically expensive to run. The authors demonstrate that standard Bayesian optimization does not work and that a specialized kernel that respects policy invariance is key to allowing Bayesian optimization to work for IRL. Experimental results show that the proposed method works on a variety of problems and is superior to other methods that only learn a point estimate of the reward function.

Strengths: Bayesian optimization is a well established field in machine learning that has not been applied directly to reward learning. This paper makes the nice contribution of combining GP Bayesian optimization with maximum entropy IRL. The proposed kernel has interesting theoretical properties and good empirical performance compared to other standard kernels. Bayesian IRL methods are typically computationally intractable due to long mixing times for MCMC and requiring an MDP solver for each proposal evaluation. BO-IRL seems like a nice middle ground between full Bayesian inference and just estimating a MLE reward function as done in most prior work. The experimental results are promising and the visualizations of the reward function space are nice. Also the paper provides nice theoretical results regarding reward shaping and policy invariance.

Weaknesses: The performance of BO-IRL is shown compared to AIRL and GCL; however, performance is not benchmarked with respect to standard Bayesian IRL. This comparison (at least for the model-based envs) should be added to really understand how this method compares to previous works. Run time results are not included. Since the proposed method requires using an MDP solver in the inner loop this seems like it could significantly slow things down compared to a GAN-style approach like GCL or AIRL. But because BO-IRL uses Bayesian optimization it may be more computationally efficient. It would be beneficial to compare each method's run-time to make a better comparison of performance/computation trade-offs.

Correctness: On line 137 the paper claims that the partition function is absent in Def 1. Isn't the denominator just an approximation of the partition function (see citations to related work below).

Clarity: The paper is well written and easy to follow.

Relation to Prior Work: It should be mentioned that using random rollouts to estimate the partition function has been done in prior work: - Boularias et al. "Relative entropy inverse reinforcement learning." AIStats. 2011. - Kalakrishnan et al. "Learning objective functions for manipulation." ICRA, 2013. - Brown et al. "Better-than-Demonstrator Imitation Learning via Automatically-Ranked Demonstrations." CoRL, 2019. . How does the current work relate to prior work on active learning for IRL? These methods typically have an aquisition function and use Bayesian reasoning/optimization to pinpoint informative queries. For example, Cohn et al.'s "Comparing action-query strategies in semi-autonomous agents" uses an acquisition function based on expected value improvement to pinpoint queries and Brown et al.'s "Risk-aware active inverse reinforcement learning," uses an acquisition function based on minimizing risk. BO-IRL is different in that it queries for reward functions to be evaluated via MaxEnt IRL but the authors claim that no one has used Bayesian optimization for IRL and the work on active learning seems very related to Bayesian optimization.

Reproducibility: Yes

Additional Feedback: Maximum entropy IRL requires full trajectories whereas Bayesian IRL can work from disjoint state-action pairs. Can you use a Bayesian optimization approach that uses state-action pairs similar to Bayesian IRL but without needing expensive MCMC sampling? Is the difference in NLL shown between Bayesian IRL and BO-IRL just a function of likelihood? Bayesian IRL doesn't use a Max Ent likelihood function, but rather has a partition function based on actions. Why do GCL and AIRL fail on the point mass maze?


Review 2

Summary and Contributions: The paper addresses the problem of inverse reinforcement learning. The main contribution of the paper is a projection operator that is invariant under potential-based shaping transformations. In other words, two rewards that are related through reward shaping using a potential-function are projected into the same point in R^n. In rough terms, given K expert trajectories, a reward is projected into a K-dimensional vector that measures the (max-entropy) likelihood of the observed trajectories in comparison with that of uniformly sampled trajectories. The proposed projection is then used within a Bayesian optimization setup to recover the reward function that maximizes the likelihood of the observed expert trajectories. The paper uses a standard Bayesian optimization setup using a Gaussian process as a surrogate model and expected improvement as acquisition function. The Gaussian process is defined over the space of projected rewards, thus leveraging the aforementioned projection operator.

Strengths: The paper a very relevant topic to the NIPS community---namely, inverse reinforcement learning. It is well-written, and the contributions proposed are elegant and, to the extent of my knowledge, novel. I really liked this paper.

Weaknesses: Perhaps the one aspect that I would like to see better discussed is the impact of the proposed projection with forms of policy invariance other than those obtained by potential-based reward shaping (more on this in the detailed comments).

Correctness: Yes, to the extent of my understanding. However, I went over the supplementary material very superficially.

Clarity: Yes.

Relation to Prior Work: Yes.

Reproducibility: Yes

Additional Feedback: I really enjoyed reading the paper. The paper is clearly written and, in my opinion, proposes an elegant solution for an important problem in IRL---the fact that the same policy may arise from multiple rewards. There are, however, a couple of aspects that I think the paper could improve upon (several of which are just minor aspects). I detail these below. I number the different issues to facilitate author's responses. 1. My main question is with respect to the use of the proposed projection with other forms of policy invariance. Towards the end of Section 3.2, the paper argues that all policy invariant reward functions should be mapped into the same point. However, I am not so convinced that this is so. For example, scaling the reward function by a positive scalar does not affect the optimal policy, but it does affect the likelihood of a trajectory in the maximum entropy approach. Therefore, I'd venture that the projection of two reward functions---one of which is a scalar multiple of the other---is not exactly the same. Am I missing something here? In any case, I think that this discussion could, perhaps, be expanded upon since, in my view at least, the projection mechanism is the key contribution of the paper. 2. Regarding the experimental section, while the results presented are compelling, I had some difficulty in parsing the posterior distribution plots, since understanding these plots requires going to the appendix. For the sake of self-containedness, I suggest adding a brief description of environments in the main text, for otherwise plots are difficult to parse. I understand that space restrictions are an issue, but perhaps Figs. 7a and 7b can be removed (since they don't add much to the discussion), and perhaps some other redundant figures could also be removed (e.g., Figs. 1 and 3 are pretty much the same...). 3. In Table 1, the ESORs marked in bold are the smaller ones. Are the rewards negative? -- Update after author's rebuttal -- Thank you for the detailed rebuttal. I am happy with the responses to the issues I've raised.


Review 3

Summary and Contributions: [UPDATE] I thank the reviewers for their response and for addressing my concerns. I'm still not convinced with the argument in lines 2-6 of the rebuttal. If s_0 is some fixed initial state, and the initial state distribution is p_0(s_0) = 1, then they would need to approximate the partition function Z(theta) in Eq. 6; so it will depend on the MDP (or problem). Thus in general it doesn't avoid the computation of Z(theta)? ==================== This paper proposes a new inverse reinforcement algorithm, named as BO-IRL, by utilizing Bayesian optimization. BO-IRL returns a posterior distribution over the reward functions that are consistent with the demonstrated behavior, and it efficiently explores the reward function space. In particular, they aim to solve the negative log-likelihood objective via Bayesian optimization. First, they illustrate the problems with directly using the standard stationary kernels for IRL. To address these issues, they propose a \rho-projection method that satisfies two “preferred” properties. Finally, they use the standard kernels on the projected space. They have compared their algorithm with baselines on synthetic and real-world environments.

Strengths: The paper proposes a novel algorithm for the IRL problem with the potential to reduce the number of exact policy evaluation steps. The paper is fairly well written, and illustrative experimental results are provided. Reproducibility: well-documented code is provided.

Weaknesses: I consider the following as the limitations of/concerns about this work: 1/ Lack of rigorous quantification of the second property of the \rho-projection (Definition 1, Eq. 6): points that are close in its range correspond to reward functions with similar L_IRL. When Eq. 6 satisfies this requirement? I guess that when rho_tau (theta) approaches p_theta (tau), this might be possible. If that is the case, the number of uniform samples M has to be large. Then, the computation of rho-projection becomes expensive (equivalent to computing the partition function). I think more discussion on the value of M is required, and better to report the values used in the experiments as well. 2/ In section 3.3, if the number of expert trajectories K is substantially larger than the dimension size of the reward parameter, does this K projection operation increase the computational complexity? 3/ They have provided a limited set of experiments to claim that the proposed method outperforms the SOTA methods. In section 3.4, they have claimed that BO-IRL can reduce the exact policy evaluations compared to ME-IRL and Deep ME-IRL. This could have been empirically demonstrated in the model-based discrete environments.

Correctness: Theoretical claims about the first property of rho-projection are correct. Experimental results are also properly reported.

Clarity: Yes.

Relation to Prior Work: Yes.

Reproducibility: Yes

Additional Feedback:


Review 4

Summary and Contributions: This paper proposes rho-projection, which is an alternative representation of a reward function in order to make a Gaussian process Bayesian optimization available on a reward function space. This paper addresses the issue of the exploration of reward parameters. This paper applies a BO technique on newly designed feature space. The proposed method is tested on three simulation environemnts and shows similar perofrmance to other compared methods in two environments and outperforms other methods in one environment. _____________________________________________________ While the response addresses some of my questions, however, there still exist unclear parts. In particular, the definition of p(tau) is still unclear. I know p(tau) is deterministic in each run and that is because tau is determined. Then, next question is whether tau is a random variable or not. To me, it is a very weird setting that tau is deterministic since most IRL methods assume that the expert shows stochastic behavior, thus, tau is usually random variables. I update my score after reading response.

Strengths: 1. theoretical grounding: weak This paper provides some theorems but, to me, all theorems are not novel. First, Corollary 1 is trivial since adding some consant value cannot change the optimal policy. Thus, Theorem 1 is not essential to prove Corollary 1. Furthermore, Theorem 1 is not novel (See Lemma 1 in [1]). Second, I think Theorem 2 is trivial. Theorem 2 has no theoretical contribution. Furthermore, Definition 1 is ill-posed. In Definition 1, F(\tau) indicates a set of uniformly sampled trajectories. Then, it means F(\tau) is a random variable? Then, rho is a random variable, too? I think using the expectation is more reasonable. Furthermore, this paper is not the first paper to define the concept of rho [2, 3]. 2. empirical evaluation: boderline Simulations show promising results of the proposed method. However, I have the following questions. 1) Why is GAIL [4] not compared? GAIL [4] is a baseline algorithm and should be compared. 2) Why do all algorithms show poor performances on the point mass maze problem? It seems the easiest one among all simulations. 3. significance and novelty of the contribution: strong I totally agree with the motiviation of this paper and the approach is quite novel. But, theoretical results and experimental results are weak to verify whether the idea is really working or not. [1] Achiam, J., Held, D., Tamar, A., & Abbeel, P. (2017, August). Constrained policy optimization. In Proceedings of the 34th International Conference on Machine Learning, 2017, (pp. 22-31). [2] Ziebart, B. D., Maas, A. L., Bagnell, J. A., & Dey, A. K. (2008, July). Maximum entropy inverse reinforcement learning. In Aaai (Vol. 8, pp. 1433-1438). [3] Ziebart, Brian D. "Modeling purposeful adaptive behavior with the principle of maximum causal entropy." (2010). [4] Ho, Jonathan, and Stefano Ermon. "Generative adversarial imitation learning." Advances in neural information processing systems. 2016.

Weaknesses: See strenghts

Correctness: Some definitions and theorems are not essential.

Clarity: This paper is generally well written.

Relation to Prior Work: Related works are well explained

Reproducibility: No

Additional Feedback:

[Author Response · NeurIPS 2020]

We thank all the reviewers for their constructive feedback. We address the key questions and concerns below.

**The role of $\rho$-projection and relationship to $p(\tau)$ (R1, R3 and R4)**: To clarify, $\rho$-projection in Eq. 6 is *not* an approximation of $p(\tau)$, despite the similar forms. $\mathcal{F}(\tau)$ in the denominator of $\rho$-projection is sampled to have the same starting point and length as $\tau$; as such, it may not cover the space of all trajectories and hence, does not approximate $Z(\theta)$ even with large $M$. The appealing property of $\rho$-projection is that the partition function is cancelled off from the numerator and denominator, thereby eliminating the need to approximate it. This is shown in Eq. 1 below.

$$\rho_\tau(\theta) \triangleq \frac{p_\theta(\tau)}{p_\theta(\tau) + \sum_{\tau' \in \mathcal{F}(\tau)} p_\theta(\tau')} = \frac{e^{R_\theta(\tau)}/Z(\theta)}{e^{R_\theta(\tau)}/Z(\theta) + \sum_{\tau' \in \mathcal{F}(\tau)} e^{R_\theta(\tau')}/Z(\theta)} = \frac{e^{R_\theta(\tau)}}{e^{R_\theta(\tau)} + \sum_{\tau' \in \mathcal{F}(\tau)} e^{R_\theta(\tau')}} \quad (1)$$

**Handling other forms of policy invariance (R2)** Scaling of reward is not a valid form of policy invariance as it changes the corresponding optimal *stochastic* policy and therefore, our projection will *correctly* map them to different points. Therefore, this is not a valid counterexample to $\rho$-projection's handling of other forms of policy invariance.

**Additional empirical comparisons (R1, R3 and R4)**: Regarding comparisons to GAIL and Deep ME-IRL: GAIL was not compared against because it is an imitation learning algorithm that retrieves policies directly and does not return a reward function. Deep ME-IRL is only applicable to the two discrete environments, out of which Borlange is infeasible due to the large state space (>20,000 states). When applied to Gridworld, Deep ME-IRL performance was very poor; it converged to $\approx$18% of expert's ESOR.

The ESOR values in Table 1 shows the number of iterations taken to reach expert's ESOR. For point mass maze, the success rates and ESOR values show the compared algorithms failed to reach the expert's ESOR *within* the set budget (100 iterations for AIRL & GCL; 50 for BO-IRL). Increasing the limit to 1000 for AIRL & GCL and 100 for BO-IRL results in higher success rates (Table 1 below). Table 1 also includes BIRL as suggested by **R1** using a) Mean of the samples collected thus far (BIRL-Mean) and b) the Policy Walk algorithm. BO-IRL with $\rho$-RBF outperforms the rest in success rates and the iterations required in most scenarios. Even though BIRL (Policy Walk) has a higher success rate in Gridworld, it comes at the cost of significantly higher iterations compared to our method. We will include the above results in the revision.

**Relationship to Active Learning (AL) in IRL (R1).** At first glance, BO-IRL might look similar to AL since they both use uncertainty measures to calculate the next query. However, they differ in the type of query used. AL queries the most informative states (actions) for additional trajectories while BO-IRL queries the likelihood of a reward function given a fixed set of demonstrations.

**Time-complexity (R3)**: $K$ only affects the calculation of covariance and is linear in complexity.

**Run-time comparison (R1)** In our experiments, the GAN-based methods currently have a faster run-time per iteration since they apply approximate policy evaluation. However, the core advantage of our method is the efficient exploration of the reward function space in order to identify multiple valid reward functions. GAN-based methods require numerous runs with random initialization to identify multiple valid reward functions which will increase their runtime.

**Significance of theoretical contributions (R4)**: Our main contribution is a Bayesian Optimization approach to IRL. We provide Theorem 2 to support our approach—Theorem 2 is important as it formalizes the key idea that the $\rho$-projection maps PBRS-based policy invariant rewards to a single point. We do not claim Theorem 1 as a contribution and we state explicitly in the paper (lines 147 and 152) that it is from [20]. We re-state the theorem and Corollary 1 in the paper for the reader's convenience.

Definition 1 is not ill-posed since $\mathcal{F}(\tau)$ is a deterministic hyper-parameter rather than of a random variable; the value of $\mathcal{F}(\tau)$ is fixed in each run rather than sampled throughout.

| Algorithm | Kernel | Gridworld | | Börlange | | Point mass maze | |
|---|---|---|---|---|---|---|---|
| | | SR | Iterations | SR | Iterations | SR | Iterations |
| BO-IRL | $\rho$-RBF | 70% | **16.0**±15.6 | **100%** | **2.0**±1.1 | **80%** | **51.4**±23.1 |
| | RBF | 50% | 30.0±34.4 | 80% | 9.5±6.3 | 20% | 28.0±4 |
| | Matérn | 60% | 22.2±12.2 | 100% | 5.6±3.8 | 20% | 56±29 |
| BIRL (Mean) | | 60% | 560.3±206.4 | 0% | - | | N.A |
| BIRL (Policy Walk) | | **80%** | 630.5±736.9 | 80% | 98±167.4 | | N.A |
| Deep ME-IRL | | 0% | — | Intractable | | | N.A |
| AIRL | | 70% | 70.4±23.1 | 100% | 80±36.3 | 80% | 90.0±70.4 |
| GCL | | 40% | 277.5±113.1 | 80% | 375±68.7 | 0% | — |

[Meta-Review · NeurIPS 2020]

Three out of four knowledgeable referees support acceptance for the contributions and I also recommend acceptance. I believe the concerns about theoretical aspects of R4 were addressed in the rebuttal. In the revised version of the paper, please present your additional experiments to address concerns of reviewers R3 and R4, your comments regarding runtime (R1) and your comments regarding the rho-projection (R1,R3,R4). Furthermore, R3 has some remaining theoretical concerns which were not clarified in the rebuttal - please elaborate on these in the revised paper.